# High-throughput engineering of a mammalian genome reveals building principles of methylation states at CG rich regions

**Arnaud R Krebs[1], Sophie Dessus-Babus[1], Lukas Burger[1,2], Dirk Schübeler[1,3]\***

[1]Friedrich Miescher Institute for Biomedical Research, Basel, Switzerland; [2]Swiss Institute of Bioinformatics, Basel, Switzerland; [3]Faculty of Science, University of Basel, Basel, Switzerland

**Abstract** The majority of mammalian promoters are CpG islands; regions of high CG density that require protection from DNA methylation to be functional. Importantly, how sequence architecture mediates this unmethylated state remains unclear. To address this question in a comprehensive manner, we developed a method to interrogate methylation states of hundreds of sequence variants inserted at the same genomic site in mouse embryonic stem cells. Using this assay, we were able to quantify the contribution of various sequence motifs towards the resulting DNA methylation state. Modeling of this comprehensive dataset revealed that CG density alone is a minor determinant of their unmethylated state. Instead, these data argue for a principal role for transcription factor binding sites, a prediction confirmed by testing synthetic mutant libraries. Taken together, these findings establish the hierarchy between the two *cis*-encoded mechanisms that define the DNA methylation state and thus the transcriptional competence of CpG islands.

**\*For correspondence:** Dirk. Schubeler@fmi.ch

**Competing interests:** The authors declare that no competing interests exist.

## Introduction

Multiple levels of regulation control correct expression level of a gene. In addition to transcription factors (TF), epigenetic signals enable temporal integration of regulatory events through dynamic processes including cell division and organism development. Considerable progress has been made in mapping the occurrence of various epigenetic marks during the course of mammalian development. This has refined our picture of the spatio-temporal occurrence of epigenetic marks, yet our mechanistic understanding on how their deposition is regulated remains limited.

In mammals, methylation of DNA occurs mainly at cytosines lying in a CG context and its presence correlates with transcriptionally repressed states (*Deaton and Bird, 2011*; *Jones, 2012*). Unmethylated CGs are concentrated in regions that are unusually rich in CG dinucleotides as compared to the rest of the genome (*Bird, 1980*). The extrapolation of this observation led to the concept of CG islands (CGI), as an operational definition of regions that are likely unmethylated based on their sequence composition. Recently generated methylation maps at basepair resolution from several tissues and organisms have experimentally identified unmethylated regions at unprecedented detail (*Hodges et al., 2011*; *Molaro et al., 2011*; *Stadler et al., 2011*; *Xie et al., 2013*; *Ziller et al., 2013*). These datasets revealed that many unmethylated regions (UMRs) extend far beyond the CGI definition and that CGI can show variable levels of methylation (*Meissner et al., 2008*; *Mohn et al., 2008*; *Doi et al., 2009*; *Hodges et al., 2011*; *Molaro et al., 2011*; *Stadler et al., 2011*). Moreover, these studies identified CG poor regions outside of CGIs having low methylation levels (Low Methylated Regions–LMRs)

**eLife digest** Regions of DNA called genes produce the proteins and other molecules that are essential for life. The act of making these molecules is known as gene expression, and being able to switch this process on and off allows cells to adapt to changing conditions. For example, some genes may be turned on in response to injury or may only turn on during waking hours.

There are several ways gene expression can be switched on and off. Proteins called transcription factors can bind to DNA and act like a switch that affects nearby genes. Alternatively, special tags called methyl groups can attach to the 'letters' that make up the DNA code and turn off gene expression. However, it is not understood how these tags work with transcription factors and other forms of gene regulation.

Regions of DNA that boost the expression of a neighboring gene are called promoters. Many promoters in mammals contain repeating patterns of the DNA letters 'C' (which is a chemical called cytosine) and 'G' (guanine), and these regions are tagged less often than other regions of DNA. This led scientists to wonder whether the DNA sequence itself controls where the tags are placed, but existing experimental techniques made it difficult to establish if DNA sequence alone can prevent tagging.

Krebs et al. created a technique that allows thousands of different DNA sequences to be inserted into the same part of the genome of mouse stem cells. Comparing the tagging across these different sequences revealed that the CG pattern is not as closely associated with tagging as was thought. If the CG pattern is repeated many times it does seem to prevent tagging, but sequences with fewer repeats also sometimes escape tagging.

Krebs et al. found that a sequence was much less likely to be tagged if the nearby DNA also contains a site that transcription factors can bind to. However, regions with a very high number of CG repeats are able to avoid tagging without help from transcription factors.

Krebs et al. found that this behavior is not seen in cancer cells. DNA in cancer cells is heavily tagged, even in CG-rich regions, and transcription factors do not appear to play a major role in directing tagging. The new approach developed by Krebs et al. should benefit researchers working to understand the multiple mechanisms that control gene activity.

(*Hodges et al., 2011*; *Stadler et al., 2011*; *Hon et al., 2013*; *Xie et al., 2013*; *Ziller et al., 2013*). These findings challenged the notion that a simple sequence definition provides the most accurate prediction for the methylation state of a DNA sequence (*Hodges et al., 2011*; *Molaro et al., 2011*; *Long et al., 2013b*) and motivated to derive improved models to predict CGIs that integrate multiple genomic features (*Bock et al., 2007*; *Wrzodek et al., 2012*; *Zheng et al., 2013*).

Several molecular mechanisms have been proposed that contribute to an unmethylated state on the basis of their correlative occurrence with unmethylated regions of the genome (*Deaton and Bird, 2011*). From sequence perspective, these include binding of unmethylated CGs by CXXC domain containing proteins, which have been proposed to inhibit or counteract methyltransferase activity directly or through the establishment of a specific chromatin state (*Ooi et al., 2007*; *Cedar and Bergman, 2009*). This model would predict that local CG content is the sole determinant of the unmethylated state. Recent experiments suggested that classical transcription factors that bind motifs more complex than CG generally promote a hypomethylated states within CG poor region (*Hodges et al., 2011*; *Stadler et al., 2011*). Notably transcription factors have been previously implicated to impact activity but also methylation state of CGI (*Brandeis et al., 1994*; *Macleod et al., 1994*; *Dickson et al., 2010*; *Lienert et al., 2011*). Yet, it is inherently difficult to dissociate the effects of CGs and TF binding sites since both coincide at UMRs. Transgenic studies have argued that both local CG concentration and binding of transcription factors (TF) have a role in promoting low methylation levels (*Brandeis et al., 1994*; *Macleod et al., 1994*; *Dickson et al., 2010*; *Lienert et al., 2011*). The cumbersome nature of targeted genetics in mammals limited the scale of these experiments preventing generalization of the observed effects as well their translation into predictive models. Nevertheless, these results demonstrate that genetic information is necessary and sufficient to instruct methylation states, opening the possibility to study the regulation of methylation states through genetic perturbation.

However this requires a large number of sequence variations in order to be comprehensive, which is a general prerequisite and experimental bottleneck in the analysis of DNA sequence contribution to regulation of biological processes. In order to be informative, such experiments have to be performed in a controlled environment to minimize context related interference. Recently such approaches have been successfully developed to dissect the organization of cis-regulatory elements by generating large pools of sequence variants and measuring their effect on transcription using transient reporter gene assays (*Melnikov et al., 2012*; *Patwardhan et al., 2012*; *Sharon et al., 2012*; *Arnold et al., 2013*). Such transient assays however are not suitable to study chromatin regulation, which requires stable genomic integration of the sequences of interest at the same chromosomal locus to account for influences of copy number and local chromatin environment.

To move beyond these limits, we developed an assay that allows parallel insertion of thousands of DNA fragments in a defined locus in murine embryonic stem cells (ESC). We isolated and synthesized various collections of DNA sequences in order to separately test the quantitative effect of sequence features proposed to influence methylation states. Parallel profiling of the methylation status of this catalogue of sequences allowed us to derive a comprehensive dataset that permits the systematic association between sequence motifs and methylation states. Using this information, we derive quantitative models describing the sequence determinants that govern the establishment of DNA methylation states. This surprisingly reveals that transcription factor binding sites are essential to explain the unmethylated state of the majority of CpG islands. We further demonstrate the utility of such datasets by explaining methylation changes observed in the course of normal differentiation. Moreover, we observe that deviation from the derived models is a characteristic hallmark of methylation changes associated with cancer states suggesting that this phenomenon is mechanistically distinct from differentiation related methylation changes.

## Results

### Recombinase based targeting of DNA libraries in murine stem cells

In order to combine genomic targeting and high-throughput measurements, we developed a method that allows parallel insertion of hundreds of DNA fragments in a defined locus in murine ESCs (*Figure 1A*). This approach entails the generation of a plasmid library of sequences of interest, which can be generated by selection or synthesis. The inserted sequences are flanked by lox sites in inverted orientation to subsequently enable *cre* mediated targeting (*Feng et al., 1999*). The library is transfected as a pool into ES cells together with an expression plasmid for the *cre* recombinase. Clones that underwent targeted exchange are selected solely based on loss of a negative selection marker at the target site. The resulting targeted cells can be analyzed through the use of universal primers flanking the fragments and subsequent sequencing. This approach yielded reproducibly up to several thousand integrants using a standardized transfection protocol (*Figure 1—figure supplement 1*). To our knowledge this generated unprecedented sequence diversity at a single genomic site in a higher eukaryote.

### DNA methylation analysis of targeted sequence libraries

CG rich unmethylated regions differ in their average CG content (*Figure 1B*). Moreover CG content varies within individual regions that contain short stretches of differential CG density (*Figure 1C*). In order to ask how this local heterogeneity translates into an unmethylated state and if different mechanisms function in different parts of islands, we inserted fragments of CGIs to define their individual potency to regulate methylation and to potentially identify DNA sequence determinants of this regulation (*Figure 1D*).

To create libraries allowing the exploration of CG rich regions, we took advantage of our previously described murine ESC methylome (*Stadler et al., 2011*). We performed an *in silico* digestion of the mouse genome using all available methylation sensitive restriction enzymes and chose a set of enzymes and defined size range of fragments to maximize the enrichment for fragments from CG rich unmethylated regions. The library was cloned and amplified in *Escherichia coli* and its composition determined by paired-end sequencing. Subsequently the libraries were targeted into a defined genomic site in murine stem cells as described above. DNA extracted from the pool of cells after selection served as a template for PCR after bisulfite conversion using universal primer binding sites that flank all inserts. The resulting PCR products were analyzed by next generation sequencing resulting in high coverage methylation measurements for ~30% of the initially inserted fragments

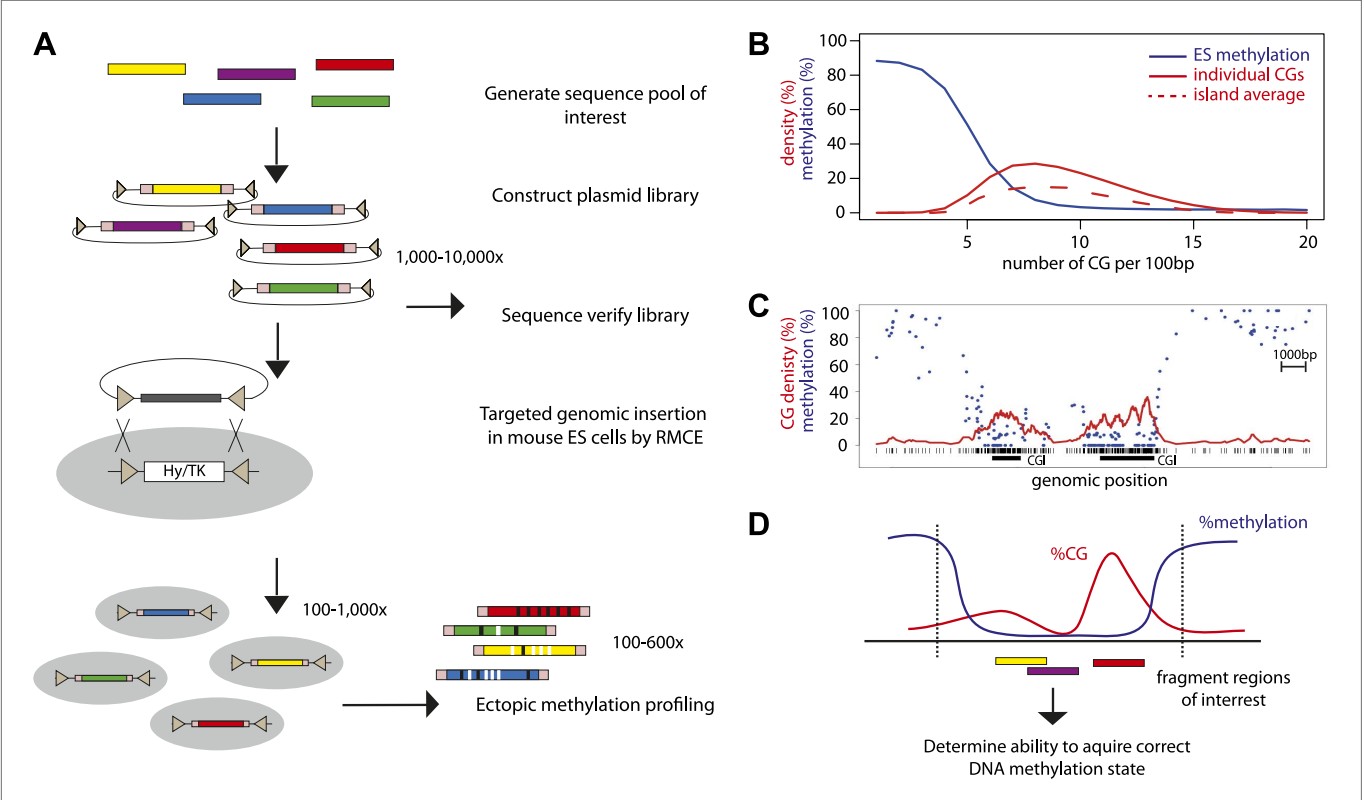

**Figure 1**. High throughput genome engineering methodology. (**A**) A pool of diverse DNA fragments is ligated into a plasmid that contains a set of inverted lox-P sites (triangles) and universal priming sequences (pink boxes) flanking the cloning site. After transformation in *E. coli* the library composition is determined by paired-end sequencing of the fragment boundaries. The plasmid library is inserted at the β-globin locus by Recombination Mediated Cassette Exchange (RMCE). Methylation of the fragments is determined by high throughput sequencing of the bisulfite PCR product produced using the universal primer sites. (**B**) Comparative distribution of methylation and densities of CG dinucleotides in the mouse genome. CGs of the mouse genome were classified based on local CG density and the average of their methylation status in mESC was plotted (blue line). The proportion of single CGs within UCSC CGIs having a certain CG density are plotted as filled red line, revealing the spread of densities observed in umethylated islands. The average CG density of islands is also plotted (dashed red line), revealing the heterogeneity between islands. (**C**) Single locus example of unmethylated CG rich regions with heterogeneous CG density. (**D**) Application of the genome engineering methodology to test the sequence contribution to methylation states. Sequences of CG rich regions are fragmented into smaller entities and the ability of theses sub-fragments to acquire methylation is assayed.

The following figure supplement is available for figure 1:

**Figure supplement 1**. Evaluation of the efficiency of the developed method.

(*Figure 1—figure supplement 1*). This translates into 100–600 fragments per transfection depending on initial library complexity.

## Density of CGs correlates with the ability of DNA fragments to recapitulate their methylation states

We initially inserted three independent libraries enriched for sequences from unmethylated CGIs. This resulted in high-resolution methylation measurements for ~400 fragments with variable length (100–400 bp). The acquisition of methylation of fragments between different insertions was reproducible for most fragments between independent library insertions (R > 0.6) (*Figure 2—figure supplement 1*). Next we compared the methylation state of fragments after insertion with that of the matching endogenous sequence (*Stadler et al., 2011*) (*Figure 2—figure supplement 1*). This analysis reveals that the majority (63.5%) of tested sequences are methylated similarly to their endogenous counterparts despite the fact that they only represent short sub-fragments of CGIs. Interestingly, the length of individual fragments did not appear to be critical for this autonomy. For example, elements as small as

100 bp were found to be sufficient to establish an unmethylated state. However 40% of the inserted fragments gained methylation relative to their endogenous site. Surprisingly, this gain of methylation does not result in a completely hypermethylated state but covers a broad range in frequency from 20–100% despite the fact that these fragments originate from regions devoid of methylation. Thus, dividing unmethylated regions into shorter entities transforms a methylation frequency that is mostly binary in the genome to a continuous variable. Having derived this observation from a large pool of sequences opens the possibility to infer quantitative relationships between sequence composition and resulting methylation.

We initially assayed how much dinucleotide frequency could explain the observed differential methylation patterns (*Figure 2—figure supplement 2*). In fact the frequency of CG explains only 14% of the observed variation (R = −0.37), suggesting that differences in CG density between fragments contribute to a certain extend to their differential methylation. A comparison of the average ectopic methylation for all fragments relative to their CG density reveals that at the upper end of CG densities (≥12CGs/100 bp) the inserted sequences tend to behave as the cognate endogenous sequence since they show little to no methylation (*Figure 2D*). Importantly however, at CG densities that are more representative for islands (*Figure 2D*, upper panel), the methylation of endogenous and inserted fragments starts to significantly deviate. While some stay unmethylated, others gain methylation as indicated by the spread of methylation levels. This trend of increased methylation over a wide range becomes stronger with reduced CG density (*Figure 2D*). Thus at very high frequencies, CGs appear sufficient to explain an unmethylated state of small fragments after insertion, while at lower densities numerous fragments of similar CG density show highly variable methylation.

## Quantitative contribution of CG densities to methylation states

CGIs represent functional regulatory regions that are under selection for the presence of transcription factor binding sites (*Deaton and Bird, 2011*; *Jones, 2012*). In order to ask how the presence or absence of such motifs affects the methylation after insertion, we next introduced libraries of sequences with different CG densities derived from the *E. coli* genome. This prokaryotic DNA is not under selection for binding sites for mammalian transcription factors, allowing us to measure the effect of CG density in an isolated context. The combination of two library designs, allowed us to measure a total of 183 prokaryotic sequences covering a broad range of CG densities. We compared their acquired methylation once inserted in the mouse genome in relation to their CG density (*Figure 3A*). Similar to the inserted mouse fragments the unmethylated state can be observed at the highest CG densities (≥12CG/100 bp) consistent with previous observations that CG density can also protect prokaryotic sequences from de novo methylation (*Lienert et al., 2011*). At a lower CG density however prokaryotic sequences get readily methylated, as revealed by higher average methylation and a significantly reduced spread compared to mouse fragments with similar CG content (*Figure 3A*). Moreover, we observe a much stronger negative correlation between CG density and methylation for these fragments than the mouse fragments (*Figure 3—figure supplement 1*, R = 0.65). The reduced spread and the high number of fragments that cover a broad range of CG densities and methylation states enable us to model the relationship between CG frequency and methylation quantitatively in a sequence context where the influence of TF sequence motifs is limited (*Figure 3—figure supplement 2*). We fitted a standard sigmoidal model, which accounts for the finite asymptotes inherent to methylation data (*Figure 3B*, see methods for complete description). The resulting graph properly describes the data ($R^2 = 0.51$).

We refer to this as the CG only model derived from prokaryotic sequences and use it to predict the methylation state of mouse fragments (*Figure 3C*) and subsequently contrast it with the actual measurement. By doing so we hope to subtract the effect of CG density on mouse fragments and furthermore isolate those that contain additional sequence cues that influence methylation. This comparison reveals that the 'CG only' model indeed has predictive power (*Figure 3C*, R = 0.38), however numerous mouse fragments are significantly less methylated than predicted based on their CG content, arguing that additional sequence cues contribute to their unmethylated state.

## Contribution of transcription factor binding

Next we wanted to ask if transcription factor binding could explain differential behavior of fragments with similar CG density. As an indirect indicator of transcription factor binding, we annotated the presence of DNaseI hypersensitive (DHS) sites in mouse ESCs at the endogenous loci from which the

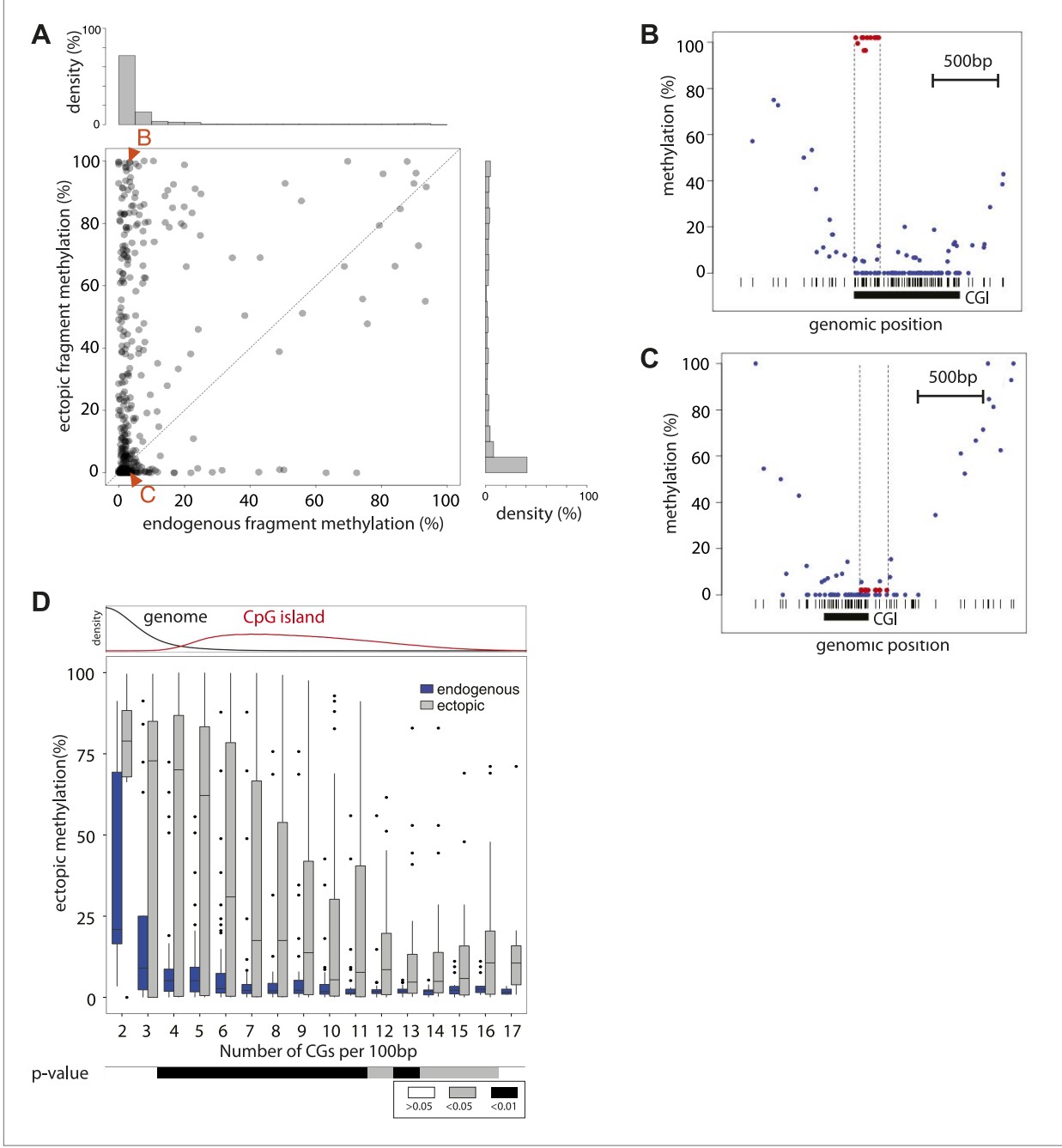

**Figure 2**. Systematic determination of the autonomy of DNA sequences to acquire DNA methylation patterns. (**A**) Comparison of methylation levels of inserted fragments with their methylation at endogenous locus (n = 394; grey transparent dots). Histograms depict the proportion of fragments in each area of the plot illustrating the prevalence of unmethylated regions within the library. A majority of these fragments maintain their state when inserted. Orange arrows indicate fragments displayed as single locus examples. (**B–C**) Examples of regions that loose or maintain their unmethyated status when inserted at the ectopic site. Single CG methylation levels for the same sequence are compared between endogenous (blue dots) and ectopic (red dots) context. Vertical lines show the boundaries of each fragment. Black box indicates UCSC CpG island definition. Black vertical bars depict the positions of CGs. (**D**) Comparison of methylation levels of DNA fragments between endogenous and ectopic context plotted against CG content. Center panel: data were binned according to the CG density of fragments and the distribution of endogenous (blue) and ectopic (grey) methylation within each bin is depicted in boxplots. Upper panel: Comparative distribution of the CG density in a 300 bp surrounding all CGs within (red) and outside (black) CpG islands throughout the genome illustrating that the vast majority of tested fragments have a CG density within the range observed at CGIs. Lower panel: statistical significance of the differences between endogenous and ectopic methylation for each CG density bin. p-values derived from a *t* test are displayed using the indicated color code for each bin.

*Figure 2. Continued on next page*

*Figure 2. Continued*

The following figure supplements are available for figure 2:

**Figure supplement 1**. Evaluation of the reproducibility of the developed method.

**Figure supplement 2**. Systematic analysis of the relationship between all dinucleotide frequency and methylation of the mouse fragments.

inserted fragments were derived (*Supplementary file 1*). Despite being predicted to be methylated based on their CG content, those fragments that are unmethylated indeed show significantly higher DHS enrichment (*Figure 3C,D*, *Figure 3—figure supplement 2*). In turn, this is compatible with the notion that TF binding motifs other than CG can contribute to the observed deviation from the CG only model.

To test this hypothesis we synthesized mutated sequences of mouse fragments that were identical in CG position and density but where all non-CG nucleotides are replaced by *E. coli* sequence in order to alter putative TF motifs. This assay was performed for fragments that we expect to have strong (*Figures 3C–1*) as well as weak (*Figures 3C–2*) protection based on their CG concentration. This reveals that those fragments for which we predict a minor role for CG density indeed display a strong shift in their methylation states while the ones where we predict CG density to have a major role only shift slightly. Thus the amplitude of methylation gain upon removal of TF binding sites is related to the CG density of the fragment, resembling the prediction from the CG only model (*Figure 3E*, R = 0.64). We conclude that in the absence of complex sequence motifs, methylation tends to approach the prediction of the 'CG only' model.

Next we asked whether insertion of TF binding motifs leads to reduced methylation in a CG rich context. We inserted both the perfect and the lowest score motif (with identical CG composition, see 'Materials and methods' for details) for the well-studied TF REST in an *E. coli* fragment that we previously observed to be fully methylated (from *Figure 3B*). Insertion of the high score motif leads to a loss of methylation while the low score motif has no effect (*Figure 3F*). Thus as previously shown for CG poor regions (*Stadler et al., 2011*), protein binding at regions with high CG densities contributes to a spatially constrained reduction of the methylation acquired by the inserted fragment.

Taken together, these results suggest that mouse fragments derived from CG rich regions are maintained unmethylated by the combined action of two different mechanisms. This argues that while CG rich regions appear homogeneously unmethylated in the genome, they are maintained at this state by cumulative effects of CG density and sequence specific protein binding.

## Modeling the relative importance of individual determinants

After assaying separately the contribution of CG dinucleotide frequency and the effect of TF binding, we tested the relative ability of each parameter to explain methylation patterns genome-wide. First we applied the 'CG only' model to predict methylation for all CGs in the genome (*Figure 4A*). Consistent with the results obtained with the inserted mouse fragments (*Figure 3C*), we found that considering only CG densities in the prediction overestimates the methylation levels for a large fraction of the CGs in the genome (*Figure 4A*, $R^2 = 0.47$, *Figure 4—figure supplement 1*). This overestimation occurs not only at CG poor low methylated regions, for which a function of transcription factors in reducing local methylation has been shown but also at CG rich UMRs. Indeed >50% of CGs within UMRs show lower methylation than predicted by their CG density (*Figure 4D*). This confirms our observations made with individual fragments and further argues for the contribution of sequences motifs other than the CG dinucleotide.

Based on our previous finding of a general link between TF binding and hypomethylation of distal regulatory regions (LMRs) (*Stadler et al., 2011*) and our experiments with mutated mouse fragments described above, we hypothesized that the remaining variance could be caused by the binding of TFs. In order to account for this effect, we attempted to define the quantitative relationship between protein binding and methylation by creating models that integrate the presence of DHS to predict hypomethylated regions. Such model indeed outperforms and complements a CG based model (*Figure 4B,C* and *Figure 4—figure supplement 1*). It improves predictions for CGs residing in LMRs as well as UMRs (*Figure 4D*), which reinforces the idea that DNA binding factors significantly contribute to the existence of unmethylated states at CG rich regions. Combining CG and DHS into one model readily

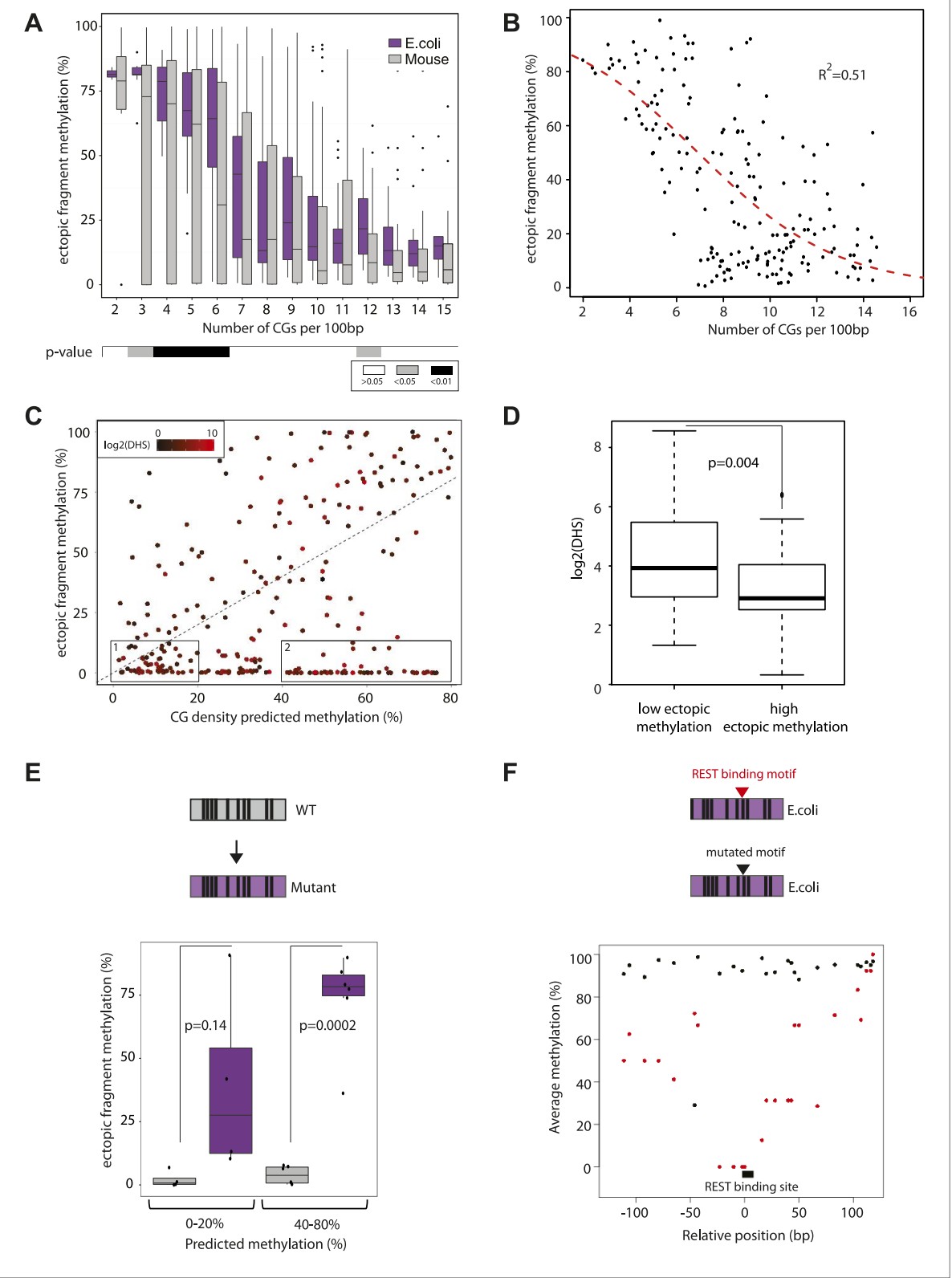

**Figure 3**. Quantification and modeling of the influence of CG content on methylation levels. (**A**) Comparison of methylation level of inserted DNA fragments derived from mouse (grey) or prokaryotic (*E. coli*-purple) genomes plotted against their CG content. Data were binned according to the CG density of the fragments and the distribution of methylation within each bin is depicted in boxplots. Lower panel: statistical significance of the

*Figure 3. Continued on next page*

*Figure 3. Continued*

differences between *E. coli* and mouse methylation for each CG density bin. p-values derived from a *t* test are displayed using the indicated color code for each bin. (**B**) Average DNA methylation levels acquired by *E. coli* DNA fragments relative to their CG density (n = 183). The methylation state of *E. coli* derived fragments is anti-correlated to its CG density. The dashed red line represents the sigmoidal model fitted to the data (Coefficient of determination of the sigmoidal fit is displayed $R^2 = -0.51$). (**C**) Comparison of the CG density based prediction and the observed methylation levels for the mouse fragments (Pearson correlation R = 0.38). The red color scale depicts the DHS signal within the fragments in their endogenous context. The two boxes show subsets used for selecting fragments to be mutated. (**D**) Boxplot comparing the endogenous DHS signal (log2 of the counts) for fragments predicted to be methylated by the CG based model and observed either unmethylated (left) or methylated (right) when inserted ectopically. The difference observed between the two groups is significant as indicated by the p-value derived from a *t* test. (**E**) Comparison of the methylation of mouse fragments to their mutated versions in which all non-CG positions were substituted by *E. coli* sequence. Boxplots for each predicted category were plotted separately for WT (grey) and mutated sequences (purple). Dots representing the methylation values for individual fragments were overlaid. Pearson correlation between the predicted value and the *E. coli* transformed is R = 0.64. p-value derived from a *t* test as a measure of statistical significance of the observed differences is displayed. (**F**) Evaluation of the effect of protein binding to methylation by insertion of a REST perfect motif (red dots), or the lowest score randomization motif (black dots) in the middle of one of the CG rich *E. coli* fragment previously found to be fully methylated.

The following figure supplements are available for figure 3:

**Figure supplement 1**. Systematic analysis of the relationship between all dinucleotide frequency and methylation of the *E.coli* fragments.

**Figure supplement 2**. Estimation of the influence of transcription factor motifs within the tested sequences.

explains over two thirds (68% genome wide and 60% at CG rich regions) of the methylation variations observed in mESCs suggesting that CG density and TF binding are the major determinants of genomic methylation.

Similar modeling in different human methylomes (*Xie et al., 2013*) revealed that these regulatory principles apply similarly in other somatic cell types and organism (*Figure 4—figure supplement 1*). Thus information derived from inserting a large number of short sequences enables to model the methylation variation observed throughout mammalian genomes. It suggests that CG density and binding of proteins to more complex transcription factor motifs explain most regions with reduced methylation in the genome.

## CG density defines methylation dynamics during cellular differentiation

Having established a predictive model consisting of two determinants to explain methylation states in mESCs, we wondered how effects driven by CG density or TF recruitments relate to the dynamics of DNA methylation observed during cellular differentiation. To do so, we compared the methylation prediction from the CG only model with the measured differences in methylation between murine stem cells and neuronal progenitors (NP). To better illustrate the influence of CG density on methylation changes, we contrasted CG rich unmethyated regions (UMRs) (*Figure 5A*), with CG poor low methylated regions (LMRs) (*Figure 5B*). As a whole CG rich regions show little variation between both cell states in line with continuous hypomethylation of CGIs during cellular differentiation. This is in contrast to CG poor regions that experience extensive methylations changes across cell lines (*Figure 5B*) (*Stadler et al., 2011*). However, detailed analysis within CG rich regions (*Figure 5A*) identifies a subgroup of cytosines that show dynamics (primarily hypermethylation) in their methylation status during differentiation. Interestingly, cytosines that change their methylation status reside within fragments where a CG only model predicts a methylated state, and little changes are observed within regions where an unmethylated state is predicted (*Figure 5A*). Notably the observed increased methylation during differentiation approaches the methylation state as predicted by the CG only model (*Figure 5A*). One likely explanation is that the unmethylated state of these sites depends on binding of TFs that are present in stem cells but not in the neuronal progenitors. Indeed, motifs for stem cell or neuron specific TFs are enriched around differentially methylated CGs (*Figure 5A*, *Figure 5—figure supplement 2*). For example we observe a methylation increase in neuronal progenitors within CG rich regions at binding sites of the stem cell specific pluripotency factor Oct4 (Pou5f1), while regions bound by factors expressed in both cell types such as REST do not change their status (*Figure 5C,D*). This effect is reminiscent of the effect of sequence mutations that abolish TF recruitment within individual fragments shown above, resulting in increased methylation that follows the CG-only model (*Figure 3D*). We conclude that variation in DNA methylation within subparts of islands is a function of TF binding and local CG density.

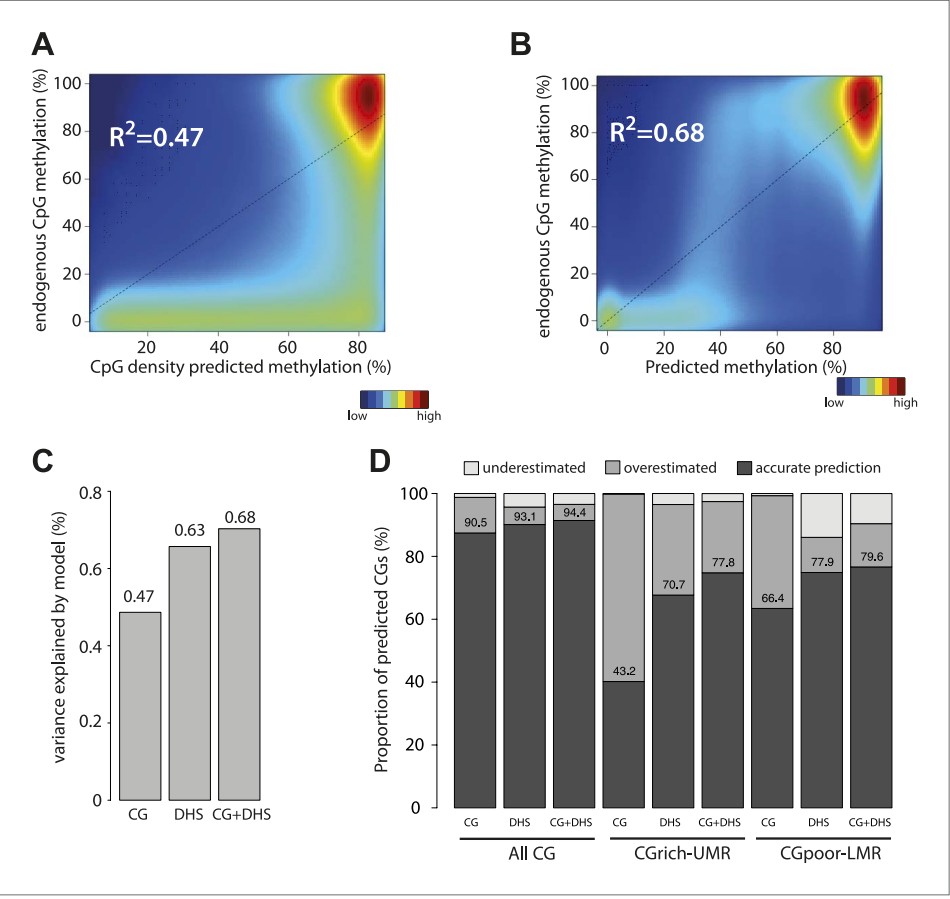

**Figure 4**. Genome wide modeling of the methylation levels combining CG density prediction and DNase hypersensitivity data. (**A**) Comparison of CG density based prediction and observed methylation levels in mouse ESC throughout the genome. Methylation is predicted using the model derived from the *E. coli* fragments. CG density is calculated in a 300 bp window around each CG in the mouse genome. The predicted value is compared to measured methylation at the single CG level in mESC. The reference line is shown in black. (**B**) Similar comparison of measured methylation at the single CG level genome wide in mESC but using a prediction model combining CG density and DHS. (**C**) Barplot comparing the explained variance by the CG only based model (CG) and the model using DHS data either alone (DHS) or in combination with CG density prediction (CG + DHS). (**D**) Proportion of CGs predicted accurately for each model relative to their genomic context. The prediction of each model was compared to methylation as measured by bisulfite sequencing and prediction accuracy was quantified (with a precision of 20% methylation). The barplot illustrates the improvement gained by each variable used in the modeling. It shows that the combination of CG density and DHS is particularly important to accurately predict methylation at CG rich regions.

The following figure supplement is available for figure 4:

**Figure supplement 1**. Performance evaluation of the derived models.

To further explore the relationship between differential protein binding and methylation differences within islands, we compared dynamics in methylation and DHS formation using an existing dataset of human ES cells and differentiated neuronal progenitors (*Xie et al., 2013*). We observe that the methylation changes are tightly anti-correlated to DHS changes at CG poor regions (*Figure 5E*). However, when DHS changes of similar amplitude are detected at higher CG densities, methylation levels do not change. This suggests that methylation levels at these regions are independent from the binding of TFs, and that a CG dependent mechanism is sufficient to explain their methylation.

These data are consistent with the model that TF binding contributes to the unmethylated state at CG poorer regions within CGIs, while TF binding is non-essential within highly CG rich regions of islands. We observe that, unlike the TF driven effect, the CG dependent effect is remarkably stable

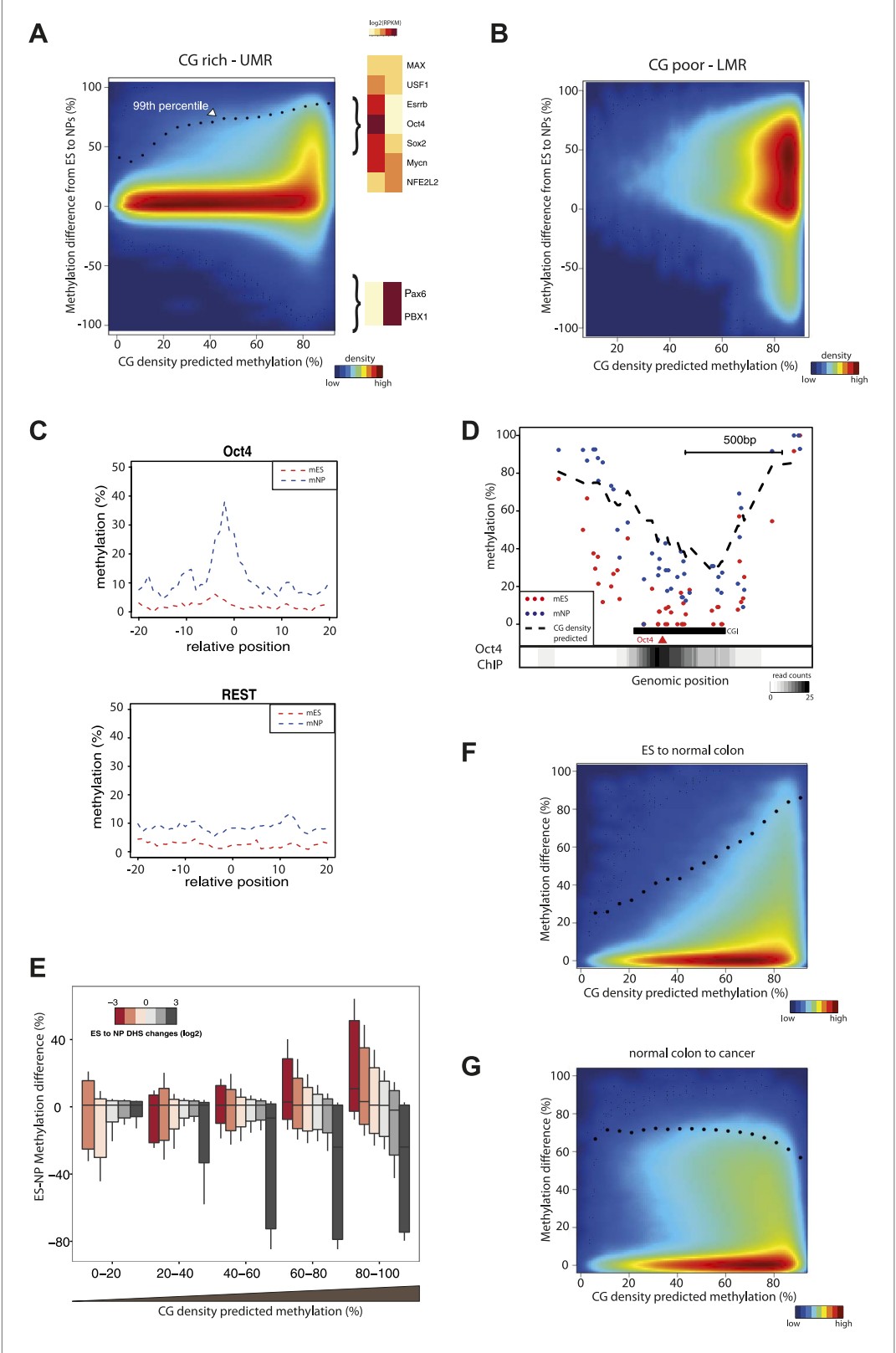

**Figure 5**. CG concentration restricts the amplitude of methylation changes during cellular differentiation. (**A–B**) Methylation gain during differentiation reaches a maximum that can be predicted by the local CG density of the region. CG density based prediction is plotted against the methylation difference between stem cells (ES) and neuronal

*Figure 5. Continued on next page*

*Figure 5. Continued*

progenitors (NP) for CGs located within stem cells (**A**) CG rich unmethylated regions (UMRs) or (**B**) low methylated regions (LMRs). Black dots represent the 99th percentile of the changes observed, as a proxy of the maximal amplitude of changes observed at a given CG density. TF motif enrichments were calculated around the most changing CGs and heatmaps depicting the expression changes for the top enriched motifs were plotted. This reveals enrichment for pluripotency factor motifs next to the hypermethylated CGs and of neuronal specific factors in the surroundings of hypomethyayed CGs. (**C**) A gain of methylation is observed at Oct4 binding sites within CG rich regions during differentiation. Composite plot depicting the average DNA methylation around Oct4 (upper panel) and REST (lower panel) binding sites at CG rich-UMRs. The red doted line represent methylation averages in embryonic stem cells (ES). The blue doted line represent methylation averages in neuronal progenitors (NP), where Oct4 is not expressed, while REST is expressed in both cell types. (**D**) Single locus example of a CG rich region that changes methylation status during differentiation, and which contains a binding site for Oct4. Lower density track represent Oct4 binding as measured by ChIP-seq (**E**) Methylation changes coincide with changes in DHS but the amplitude of change is limited by CG density. CGs present at UMRs were classified based on the CG density of their surroundings (x-axis) and the amplitude of DHS changes from ES to NP (box plot color correspond to log2(delta DHS)). The distribution of methyla- tion changes (y-axes) is depicted by a box plot for each category (methylation difference %). The figure illustrates that DHS changes correlate with methylation changes, this correlation is lost at very CG rich stretches. (**F–G**) Cancer related hypermethylation is not restricted by CG density. CG density based prediction is plotted against methylation differences between (**F**) hES cells and normal colon, (**G**) normal colon and cancer colon for CGs located within hES CG rich unmethylated regions. Black dots represent the 99th percentile of the changes observed.

The following figure supplements are available for figure 5:

**Figure supplement 1**. Characterization of the sub-regions within CpG islands that are independent from transcrip- tion factors for being unmethylated.

**Figure supplement 2**. Motif enrichment analysis at differentially methylated regions within CpG islands.

**Figure supplement 3**. Hypermethylation at CGs with low predicted methylation is a widespread and specific mark of cancer.

---

across cell types. One potential explanation is that the responsible trans-acting factors are widely expressed, as it is the case for some of the CXXC domain containing proteins, which are *bona fide* CG binders (*Long et al., 2013a*). A direct consequence of this observation is that within islands, a subset of CGs embedded in short CG dense patches are unmethylated regardless of the binding by TFs in their neighborhood (*Figure 5—figure supplement 1*). Importantly however, these patches are not essential to form unmethylated regions, as they are absent in 50% of all CGIs (*Figure 5—figure supplement 1*) and only contain 30% of all CGs within islands. They further do not show any partic- ular positioning pattern within islands (*Figure 5—figure supplement 1*). This suggests that for the majority of CGs (67.5%) within islands TF binding could contribute to their unmethylated state.

## Cancer related hypermethylation is not restricted by CG density

Our data suggest that methylation dynamics observed within islands during differentiation are mostly the result of changes in the binding profiles of transcription factors. Additionally, we observed that the local concentration of CGs restricts the amplitude of these TF driven changes. Aberrant methylation, including hyper-methylation of CGIs is an established hallmark of cancer (*Baylin and Jones, 2011*). Since the origin and the regulation of these changes during transformation are largely unknown, we wondered how these changes relate to our model.

To this end, we contrasted the methylation gain at UMRs between human stem cells (hES) and nor- mal colon cells and between normal colon and related cancer cells (*Berman et al., 2012*). This reveals that normal colon cells show methylation changes that follow our model, while colon cancer cells dis- play a distinct hyper-methylation phenotype (*Figure 5F,G*). We observe a significant gain in methyla- tion even at CGs that are constitutively unmethylated in normal tissues due to their high CG density (*Figure 5F,G*). Interestingly, the same is observed when comparing a large set of healthy tissues and cancer types arguing that this is not unique to colon cancers (*Figure 5—figure supplement 3*). This argues that cancer hyper-methylation at islands is mechanistically distinct as it cannot be modeled with the same local determinant that describe methylation changes during normal development. Since

both CG driven and TF mediated effects are affected this can unlikely be explained by alterations in transcription factor binding patterns.

## Discussion

In this study, we report a high-throughput genome engineering protocol and demonstrate its potential to determine the contribution of DNA sequence to the establishment of epigenetic states. We establish that DNA insertions at a given locus in mammalian genomes through RMCE can be performed with high complexity DNA libraries, opening the possibility to dissect regulatory mechanisms such as the ones that govern the establishment of DNA methylation patterns. In principle this approach can be adapted for any genomic readout (e.g. transcription or replication) in order to understand the interplay between multiple regulatory layers that function in *cis*. It further circumvents the limitations of previously used transient transfections (*Melnikov et al., 2012*; *Patwardhan et al., 2012*; *Sharon et al., 2012*; *Arnold et al., 2013*), which lack a proper chromosomal context and are not controlled for amount of sequence variants per cell and their copy number.

We measured the ability of several thousand DNA sequence variants to acquire DNA methylation in mouse embryonic stem cells. The specific design and scale of the tested libraries enabled us to gain insights into the sequence determinants that drive the establishment of unmethylated states at CG rich regions and to build predictive models based on these methylation measurements and the underlying DNA sequence characteristics.

One of our key findings is the quantification of the effect of CG density on methylation states. Surprisingly, while we confirm that CG density is sufficient to create an unmethylated state, only every second island harbors sequence stretches above that threshold (*Figure 5—figure supplement 1*). Combined, these CG rich stretches only cover 30% of the total number of CGs that reside in islands. Mechanistically, it remains to be determined how CG richness drives unmethylated states, yet several scenarios are compatible with our observations. Proteins that specifically recognize unmethylated CG via a CXXC domain could antagonize de novo DNA methylation as previously proposed (*Long et al., 2013a*). Alternatively, presence of dinucleotide concentrations such as CG stretches could alter nucleosomal organization (*Brogaard et al., 2012*; *Struhl and Segal, 2013*) and thereby impact DNA methylation. Since, transcription factors and CG richness can both impact nucleosome positioning (*Struhl and Segal, 2013*), it is tempting to speculate that nucleosome depletion could be a unifying principle underlying the formation of unmethylated regions.

Our data suggest that a key driver of hypomethylated states is the local binding of transcription factors. Others and we have recently shown that TF binding creates reduced methylation at CG poor regulatory regions such as tissue specific enhancers (*Hodges et al., 2011*; *Stadler et al., 2011*), a finding that extends to CG rich regions. Their methylation however is less dynamic during cellular differentiation since most CGIs are constitutively active as promoters of housekeeping genes. Moreover the effect driven by CG density is stable across cell types restricting the amplitude of the changes. Our data argue that the higher the CG content, the lower the TF dependent changes in methylation and vice versa.

This scenario explains mechanistically why unmethylated regions frequently extend beyond CGI definitions (*Hodges et al., 2011*; *Molaro et al., 2011*; *Long et al., 2013b*) and predict that the methylation changes at borders of CG rich regions (also referred to as CGI shores (*Doi et al., 2009*)) are a function of TF binding. Importantly however the observed variability in methylation is not restricted to any particular location within CGIs but is only defined by local CG frequency. We believe it is important to account for this property when studying differential methylation patterns.

We identified a striking contrast in the type of CGs affected by methylation changes within CG rich regions during normal differentiation vs those that occur during transformation to a cancerous state. Notably it is not individual cytosines that are predictive for cancer, as these vary widely between types, but it is rather the class of CGs affected as defined by our CG density model. While the nature of this predictive difference remains unknown, we note that unlike methylation changes occurring during normal cellular changes, the loss of TF binding is unlikely to explain the observed differences. Hypermethylation in cancer particularly targets CGIs that control genes that are inactive (*Gebhard et al., 2010*; *Berman et al., 2012*; *Sproul et al., 2012*). Moreover, the study of cancer methylomes revealed that hypermethylated CGIs were embedded within larger domains of intermediate methylation (Partially Methylated Domains–PMDs) (*Hansen et al., 2011*; *Berman et al., 2012*). If CGI methylation changes within PMD's are a general phenomenon it might indicate that the observed changes at CGIs in cancer reflect the loss of the local sequence autonomy in determining correct DNA methylation.

The observation that TF binding contributes substantially to hypomethylation at CGIs has potential implications for their evolutionary origin as it is compatible with the idea that the emergence of CG rich regions could have occurred indirectly through TF-driven demethylation and resulting reduced C to T transition. This is in line with reports of limited positive selection for the CG content of islands (*Cohen et al., 2011*; *Molaro et al., 2011*) and that nucleotide composition including CG frequencies vary substantially between unmethylated regions across vertebrates (*Long et al., 2013b*). Thus our findings support the notion that CGIs arose as an evolutionary footprint of ancient regulatory regions. Such scenario is still compatible with a subsequent specialization of proteins such as CFP1 or KDM2A to recognize unmethylated CGs by CXXC domains in order to target chromatin processes to regulatory regions (*Blackledge et al., 2010*; *Thomson et al., 2010*).

## Materials and methods

### Library cloning

For targeted insertion, DNA libraries were cloned into a plasmid containing a multiple cloning site flanked by priming regions for a pair of universal primers and two inverted L1 *Lox* sites (pL1-LPP1-1L).

For constructing the mouse primary libraries, a set of enzymes was selected on the basis of a screen of all available CG methyl-sensitive enzymes (NEB - Ipswich, MA). An *in silico* digest of the mouse genome was performed masking all methylated regions in ESCs (*Stadler et al., 2011*), and multiple size selection were tested to optimize the enrichment in CG rich unmethylated regions and the number of unique fragments isolated. Three enzymes (NarI, BstUI, BssHII) were selected on the prediction that they would produce a complex library (>1000 fragments) of which over 80% of the fragments would overlap with CGIs.

100 µg of mES cells gDNA (background: ES 129S6/SvEvTac) was digested and resulting fragmented DNA was size selected (100–600 bp) on a 1% agarose gel. The isolated DNA inserts were directly cloned in the receiving vector (L1-LPP1-L1). The plasmid pool was transformed and amplified in XL1-competent cells. Library complexity was estimated based on size diversity of colony PCR products prior composition determination by sequencing.

For prokaryotic libraries, a similar approach was employed, digesting *E. coli* DNA (NC_010473.1) with MspI and size selecting fragments (100–600 bp). Additionally, to be able to cover the lower and higher extremes of CG densities in a focused library, a PCR based library was cloned. 96 pairs of primers were *in silico* designed to target these regions. After PCR amplification, the products were pooled and cloned in the receiving vector.

For synthetic libraries, the sequence was designed *in silico*, custom synthesized, sequence verified (IDTechnologies, Coralville, IA or GeneArt, Life Technologies, Carlsbad, CA) and cloned in the receiving vector.

### Library insertion

The Recombinase-mediated Cassette Exchange (RMCE) insertion protocol (*Feng et al., 1999*; *Lienert et al., 2011*) was refined in order to scale the needs of inserting large number of fragments in parallel. Briefly, TC-1 ES cells were selected under hygromycin (250 µg/ml, Roche, Switzerland) for 10 days. Next, $12 \times 10^6$ cells were electroporated (Amaxa nucleofection, Lonza, Switzerland) with 75 µg of L1-library-1L plasmid and 45 µg of pIC-Cre. Negative selection with 3 µM Ganciclovir (Roche, Switzerland) was started 2 days after transfection and continued for 10 days. Pools of selected cells were tested for successful insertion of DNA libraries by PCR using primers recognizing the universal priming region flanking the insertion site.

### Determination of the composition of libraries

Direct sequencing of the fragments ends by paired end sequencing was used to determine the sequence composition of the DNA libraries derived by restriction digest of the mouse and prokaryotic genomes. To this end, the library containing plasmids were used as a template for 15 cycles of PCR using the universal set of primers flanking the fragment insertion site. The purified product was then used for standard Illumina library preparation and sequenced on a MiSeq instrument (Illumina, San Diego, CA).

Reads were aligned against the corresponding reference genome using Bowtie (*Langmead et al., 2009*) and fragments identity was called using read pairing information. A reference set of regions was established where only fragments without overlaps within the library were retained (to avoid ambiguous read assignments during methylation call of sonicated material).

## Insertion rate determination

A similar procedure than above was used to call genomic insertion rates. PCR was performed with primers annealing to the non bisulfite-converted DNA (5'-CCAACCTGACTGTGGTGGACAA-3; 5'-ACATGCACCTTCCCAGGGC-3'). The product was sonicated, gel purified and a sequencing library was prepared.

## Synthetic fragments design

Generation of sequence mutants of the mouse fragments: Fragments were selected based on their high (>40%) or low (<20%) predicted methylation according to their CG density. For each fragment all non CG positions of the sequence were replaced using a non CG containing stretch of DNA from *E. coli* as a template.

Generation of TF motif insertions: The receiving cassette was derived from an *E. coli* fragment observed to be methylated in the *E. coli* library with a CG density typical for CGI. A BamHI-XbaI entry site was *in silico* inserted in the middle of the fragment and fragment was synthetized. REST position weight matrix was extracted from the JASPAR database (*Portales-Casamar et al., 2010*) and was used to derive the best score motif (most frequent base at each position of the PWM– GACTTTCAGCACCATGGACAGCGCCACTG); and the lowest score motif (lowest score randomized motif with identical base composition and CG content–CCTCAGGTTGGCACACCTCTAAGAGCCGA). These sequences were used to synthetize pairs of oligonucleotides with flanking 5'and 3'sequences to form sticky ends for BamHI, XbaI respectively (CTAG-5'; 3'-GATC). Oligonucleotides were annealed and cloned into the receiving cassette.

## Methylation profiling

Genomic DNA (2 µg) of ES cells carrying the libraries was bisulfite converted with the EpiTec Bisulfite Kit (QIAGEN, Germantown, MD). Libraries were amplified by PCR (AmpliTaq Gold Life Technologies, Carlsbad, CA) using bisulfite compatible primers (5'-AACCTAACTATAATAAACAACC-3'; 5'-GGTATATGTATTTTTTTAGGGT-3') annealing to the universal priming region flanking the fragments cloning site. The PCR product was gel purified and fragmented by sonication (Covaris S220, Woburn, MA). The sonicated material was used to construct sequencing libraries following Illumina's recommendations. Samples were sequenced as barcoded pools on Illumina GAII or MiSeq instruments.

## Data-processing

Bismark/Bowtie 0.12.7 (*Langmead et al., 2009*; *Krueger and Andrews, 2011*) were used to align bisulfite reads against an *in silico* converted reference genome (C > T and G > A) and call methylation state for each CG. Only CGs covered by at least 10 reads were used for analysis. Strain specific SNPs were masked. Methylation was called per CG and fragment averages were derived using the previously established reference set of regions for the library. Only fragments where >50% of the CG and a minimum of four CGs were covered were considered in the analysis.

Bowtie 0.12.7 was used for aligning the non-bisulfite reads from native PCR experiments used to call insertion rates. Fragments were called inserted when >50% of the fragment sequenced was covered by the reads.

## Modeling

Data modeling was conducted stepwise by first integrating information from prokaryotic insertions, and then combining it with mouse genomic data.

For CG content analysis, only fragments ≥250 bp were considered in order to avoid scoring instability while size normalizing low CG counts. A sigmoidal model was fitted to the prokaryotic data describing the relationship of methylation to CG density at the level of fragments averages.

$$y = \frac{100}{1 + e^{-b(x-c)}}$$

Both higher and lower asymptotes were fixed prior model fitting (100% and 0% methylation) since these are known in the case of DNA methylation data. The best-fit model was then retained (b = −0.337, c = 6.917). Note that a linear model was also tested and performed equally well on the linear part of the data, however the sigmoidal model out performs it to describe both the lower and higher ends of CG densities.

Coefficients derived from this model fitted on the prokaryotic data were then used to predict methylation of (1) the mouse fragments (2) all CGs genome wide (considering a 300 bp window around each CG for CG density calculation).

In the second step of modeling, a linear model was used to combine the prokaryotic based model and transcription factor binding information as measured by DNAse-seq. The model inputs were the single CG mESC methylation levels, prediction of methylation for each CG of the genome based on the prokaryotic model and DNAse-cuts collected in a 300 bp window surrounding the CG. Prior to regression, DNAse-seq data were pre-processed to remove outliers and categorize the data. Fitting of the models were conducted on two chromosomes and performance was assessed on the rest of the genome using R-squared values.

For the analysis of methylation dynamics, segments (UMR, LMR, FMR, PMDs) were called using MethylSeekR (*Burger et al., 2013*) in the different cell types. Then single CG average methylation was compared between the two cell types in each segment type.

## Acknowledgements

The authors thank Gaidatzis Dimosthenis for help on implementing the modeling, Matthew Lorincz, Michael Stadler and members of the Schübeler laboratory for feedback on the manuscript. ARK is supported by an EMBO long-term postdoctoral fellowship. Research in the laboratory of DS is supported by the Novartis Research Foundation, the European Union (NoE 'EpiGeneSys' FP7-HEALTH-2010-257082, 'CADMAD' FP7-ICT-265505 and the 'Blueprint' consortium FP7-282510), the European Research Council (EpiGePlas), the SNF Sinergia program, and the Swiss initiative in Systems Biology (RTD Cell Plasticity).

## Additional information

### Funding

| Funder | Grant reference number | Author |
| --- | --- | --- |
| EMBO | Long Term Fellowship | Arnaud R Krebs |
| Novartis | | Arnaud R Krebs, Sophie Dessus-Babus, Lukas Burger, Dirk Schübeler |
| European Union | NoE 'EpiGeneSys' FP7-HEALTH-2010-257082, 'CADMAD' FP7-ICT-265505, 'Blueprint'; FP7-282510 | Dirk Schübeler |
| European Research Council | EpiGePlas | Dirk Schübeler |
| Swiss Initiative in Systems Biology (SystemsX.ch) | RTD Cell Plasticity | Arnaud R Krebs, Sophie Dessus-Babus, Lukas Burger, Dirk Schübeler |
| Swiss National Science Foundation | Sinergia Program | Dirk Schübeler |

The funders had no role in study design, data collection and interpretation, or the decision to submit the work for publication.

### Author contributions

ARK, Conception and design, Acquisition of data, Analysis and interpretation of data, Drafting or revising the article; SD-B, Protocol optimization and sample preparation for sequencing, Acquisition of data; LB, Analysis and interpretation of data, Drafting or revising the article; DS, Designed the study, wrote manuscript, Conception and design, Drafting or revising the article

## Additional files

### Supplementary file
• Supplementary file 1. List of external datasets used in this study.

## Major dataset

The following dataset was generated:

| Author(s) | Year | Dataset title | Dataset ID and/or URL | Database, license, and accessibility information |
|---|---|---|---|---|
| Schuebeler D, Krebs A | 2013 | Identification of building principles of methylation states at CG rich regions by high-throughput editing of a mammalian genome | GSE51170; http://www.ncbi.nlm.nih.gov/geo/query/acc.cgi?acc=GSE51170 | Publicly available at GEO (http://www.ncbi.nlm.nih.gov/geo/). |

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
