## [Decision Letter]

Thank you for sending your work entitled “High-throughput engineering of a mammalian genome reveals building principles of methylation states at CG rich regions” for consideration at *eLife.* Your article has been favorably evaluated by Chris Ponting (Senior editor), a Reviewing editor, and 2 reviewers.

All involved agree that the volume of work is impressive, as is the sophistication of the analysis. One of the reviewers had no suggestions for improvement. The Reviewing editor and other reviewer have made some suggestions for improvement and we would like you to modify the text of the manuscript accordingly.

1) The claims in the Abstract are a bit too general. “Modeling of this dataset revealed that CG density alone is a minor determinant of the unmethylated state of CpG islands. Instead, these data identify a principal role for transcription factors, a prediction confirmed by testing synthetic mutant libraries.” The only TF tested in any detail is REST, with one Figure panel pertaining to Oct4 and no studies were done using protein binding or other functional assays, so it is based on presence of binding sites only. While DHS sites are also used, these are not only determined by TFs.

2) The authors make no mention of island length, which is also highly variable. Tested fragments that are much smaller than their host CGI might be expected to fail to recapitulate the methylation state of the whole island when moved to an ectopic location, as indeed they do. With regard to the model, an R2 value of 0.51 for the sigmoidal curve is not great and visual inspection would support this view, since the curve misses large clusters of points in *E. coli* (e.g. between 8-15 below 20% in Figure 3). A steeper curve might fit better here and would also fit with biological observations (low CG density low meth and vice versa, with a steep transit in middle). It is perhaps not surprising then that many mouse fragments do not follow this model. I wouldn't agree that “the influence of TF sequence motifs is limited” in *E. coli* sequence: perhaps eukaryotic TFs, but the influence of prokaryotic TFs is still likely to be high.

3) Some of the interesting conclusions in the Discussion are not easy to relate to the data, e.g. “only every second island harbors stretches above the needed threshold...these only cover 30% of the total # CGs that reside in islands”: where do these figures come from? Or “the observed variability in methylation is not restricted to any particular location within CGIs”: the authors don't look at this effect across CGI in situ.

---

## [Author Response]

*1) The claims in the Abstract are a bit too general. “Modeling of this dataset revealed that CG density alone is a minor determinant of the unmethylated state of CpG islands. Instead, these data identify a principal role for transcription factors, a prediction confirmed by testing synthetic mutant libraries.” The only TF tested in any detail is REST, with one Figure panel pertaining to Oct4 and no studies were done using protein binding or other functional assays, so it is based on presence of binding sites only. While DHS sites are also used, these are not only determined by TFs*.

The reviewers seem to imply that there are DHS that are not TF dependent. While one cannot rule out that TF independent DHS signal exists, many lines of evidence argues that TF binding is a dominant component of the DHS. Nevertheless, we do agree that we do not test factors but rather the presence/absence of their binding sites. Thus we changed the Abstract to read “Instead, these data identify a principal role for transcription factor binding sites…” We hope this solves this point.

*2) The authors make no mention of island length, which is also highly variable. Tested fragments that are much smaller than their host CGI might be expected to fail to recapitulate the methylation state of the whole island when moved to an ectopic location, as indeed they do*.

The length of the fragments did not appear to be a significant explanation for the methylation at the ectopic site. While we do not provide a figure highlighting this point, this was stated in the original version of the manuscript: ‘…the length of individual fragments did not appear to be critical for this autonomy. For example, elements as small as 100bp were found to be sufficient to establish an unmethylated state.’

*With regard to the model, an R2 value of 0.51 for the sigmoidal curve is not great and visual inspection would support this view, since the curve misses large clusters of points in* E. coli *(e.g. between 8-15 below 20% in*
Figure 3*). A steeper curve might fit better here and would also fit with biological observations (low CG density low meth and vice versa, with a steep transit in middle). It is perhaps not surprising then that many mouse fragments do not follow this model. I wouldn't agree that “the influence of TF sequence motifs is limited” in* E. coli *sequence: perhaps eukaryotic TFs, but the influence of prokaryotic TFs is still likely to be high*.

We have tested several models and the presented sigmoidal fit is slightly outperforming a linear model as mentioned in the text. We believe it correctly describes the data and thus its used is justified in this context.

The comment regarding prokaryotic TFs motifs is unclear to us. We introduce the *E. coli* sequences in a mouse cell line were prokaryotic TFs are not present. Thus we think it is a fair assumption to state that influence of TFs is limited over those sequences.

*3) Some of the interesting conclusions in the Discussion are not easy to relate to the data, e.g. “only every second island harbors stretches above the needed threshold...these only cover 30% of the total # CGs that reside in islands”: where do these figures come from? Or “the observed variability in methylation is not restricted to any particular location within CGIs”: the authors don't look at this effect across CGI in situ*.

These statements relate to panels of Figure 5—figure supplement 1. These were not adequately referenced in the text. We have now improved the referencing of these figures in the Results section and within the Discussion. We hope this improves clarity.